# Template-Free Manufacturing of Defined Structure and Size Polymeric Microparticles

**DOI:** 10.3390/nano13222976

**Published:** 2023-11-20

**Authors:** Arkady S. Abdurashitov, Pavel I. Proshin, Gleb B. Sukhorukov

**Affiliations:** 1Vladimir Zelman Center for Neurobiology and Brain Rehabilitation, Skolkovo Institute of Science and Technology, Bolshoy Boulevard 30, p.1, 121205 Moscow, Russia; pavel.proshin@skoltech.ru; 2School of Engineering and Materials Science, Queen Mary University of London, Mile End Road, London E1 4NS, UK; 3Life Improvement by Future Technologies (LIFT) Center, 143025 Moscow, Russia

**Keywords:** microparticles, scalable manufacturing, laser cutting, drug delivery, complex structures, composites, core-shell, defined size

## Abstract

Complex-structured polymeric microparticles hold significant promise as an advance in next-generation medicine mostly due to demand from developing targeted drug delivery. However, the conventional methods for producing these microparticles of defined size, shape, and sophisticated composition often face challenges in scalability, reliance on specialized components such as micro-patterned templates, or limited control over particle size distribution and cargo (functional payload) release kinetics. In this study, we introduce a novel and reliably scalable approach for manufacturing microparticles of defined structures and sizes with variable parameters. The concept behind this method involves the deposition of a specific number of polymer layers on a substrate with low surface energy. Each layer can serve as either the carrier for cargo or a programmable shell-former with predefined permeability. Subsequently, this layered structure is precisely cut into desired-size blanks (particle precursors) using a laser. The manufacturing process is completed by applying heat to the substrate, which results in sealing the edges of the blanks. The combination of the high surface tension of the molten polymer and the low surface energy of the substrate enables the formation of discrete particles, each possessing semi-spherical or other designed geometries determined by their internal composition. Such anisotropic microparticles are envisaged to have versatile applications.

## 1. Introduction

The production of polymeric microparticles of defined sizes and shapes plays a pivotal role in various industries, encompassing pharmaceuticals, biotechnology, and materials science [1]. These microparticles serve essential functions in drug delivery, diagnostics, and as functional materials [2,3]. Prolonged delivery of drugs and therapy for various diseases are the most challenging and sought-after tasks of encapsulation. Although short-term drug dosage is well covered by gel-based microcapsules [4,5,6], long-term sustained drug deposition is not a trivial task, the solution of which requires both minimizing the initial burst release and maintaining the required concentration of the active substance over a long period of time (weeks).

Numerous techniques have been proposed for manufacturing polymeric microparticles capable of long-term elution. Obviously, each of these techniques has advantages and limitations. Broadly, these methods of shape-controlled particle fabrication can be categorized into two main groups: template-free and template-based. It is worth noting that there are unique methods for manufacturing particles and other containers using additive 3D printing technologies that do not fall into one of the above-mentioned categories [7,8].

Template-free approaches primarily rely on the surface tension forces of polymer solutions or suspensions to create spherical-like structures, while template-based methods necessitate the use of specially crafted molds with surface geometries that dictate the shape and size of the microparticles. In this concise introduction, we will explore some, though not an exhaustive list, of the common and well-established techniques for producing polymeric microparticles.

### 1.1. Emulsion/Solvent Evaporation

This is one of the most widely used methods for microparticle production [9,10,11]. It involves dissolving a polymer in an organic solvent and then emulsifying it in an aqueous phase. Microparticles are formed as the solvent evaporates, leaving behind solid polymer particles.

*Pros:* allows control over microparticle size and morphology; suitable for both hydrophilic and hydrophobic polymers; can encapsulate a variety of payloads, including drugs and nanoparticles.

*Cons:* use of surfactants that can be hard to remove; encapsulation efficiency may vary; batch-to-batch variability can be an issue.

### 1.2. Spray Drying/Electrospray/Electrospinning

A polymer solution or suspension is atomized into fine droplets or thin fibers and dried rapidly either using mechanical energy like ultrasound or electrical field energy [12,13,14,15,16]. A core-shell variant of such techniques allows the increase of encapsulation efficiency. Defined sizes of microcapsules can be achieved by varying the energy of the atomization and the amount of the polymer in the solution/suspension. In the case of electrospinning, the fibers are collected as a nonwoven mat or chopped into microparticles. It offers a high surface area and is used in tissue engineering and filtration applications.

*Pros:* rapid and continuous production; high encapsulation efficiency for certain materials in the core-shell variant; suitable for thermally sensitive compounds; scalable for medium to large-scale production.

*Cons:* limited control over particle/fiber size distribution; substance-dependent encapsulation efficiency; it can be sensitive to environmental factors, which might affect the repeatability of the resulted particles/fibers; requires specialized and dangerous equipment, like high voltage sources or high-energy ultrasonic atomizers.

### 1.3. Microfluidics

Microfluidic devices can precisely control fluid flow to generate monodisperse microparticles. Continuous and highly controlled production is possible. The resulting particles are suited for drug delivery and device calibration purposes [17,18,19].

*Pros:* high precision and control over microparticle characteristics; allows for the creation of multifunctional microparticles; suitable for encapsulating delicate payloads.

*Cons:* requires microfabrication expertise and equipment; complex setup and maintenance; limited to low production rates; potential clogging issues in microfluidic channels.

### 1.4. Template-Based Methods

These methods usually utilize hard or semi-hard templates with defined geometry of a given size and shape to form polymeric microparticles. In some approaches, microstructures like vaterite particles, micelles, or other micro particulates can serve as a template on which polymer coating could be deposited by the electrostatic or other bonding forces [20,21,22,23,24,25].

*Pros:* precise control over microparticle size and shape; can create complex and non-spherical microparticles; versatile, with various template materials and fabrication techniques available; high encapsulation efficiency.

*Cons:* templates can be challenging to remove for specific geometries; production can be time-consuming, especially for intricate designs; template fabrication may require specialized equipment.

The manufacture of polymeric microparticles is a critical step in various applications, and the choice of method depends on specific requirements. Emulsion-solvent evaporation and spray drying offer simplicity and scalability but may lack precision. Template-based methods combine control with versatility but involve template removal challenges and scalability issues. Each method has its own set of advantages and limitations, highlighting the importance of selecting the most suitable technique based on the desired microparticle characteristics and production scale. The aim of this study was to develop a novel method for manufacturing particles on low-surface-energy substrates (18–20 dynes/cm). At its core, this proposed technique combines the simplicity and natural occurrence of microcapsule formation driven by surface tension forces with precise control over their size and shapes in a deterministic manner.

## 2. Materials and Methods

The following components were obtained from Sigma-Aldrich, St. Louis, MO, USA: Polycaprolactone (PCL, M_w_ = 80,000 Da), Polyvinylpyrrolidone (PVP, M_w_ = 55,000 Da), Methylene Blue, Oil Red O, Ethyl Acetate. Fluorinated Ethylene Propylene Copolymer sheets (FEP, thickness 100 µm) were purchased from PLASTPOLYMER company (St Petersburg, Russia). Trichloromethane (TCM, chloroform) was ordered from EKOS.

### 2.1. Solutions Preparation

A set of blank (cargo-free) solutions was prepared as follows: 0.19 g of PCL was added into 2.98 g of TCM. In addition to that, a mixture was stained using 0.005 g of Oil Red O to allow for laser cutting. Solutions were vigorously stirred (1500 rpm) for 2 h until complete dissolution of their components. To prepare solutions containing Methylene Blue, its crystals were first grounded in a ball mill for an hour to reduce their average size. The volume of balls was roughly twice the volume of cargo crystals. Crystal grinding was stopped when the average crystal size reached less than 1 micrometer. Fine dye powder was added into the blank solutions in quantities equal to 5, 10, and 15% of the PCL mass and dispersed using an ultrasonic homogenizer. To make a solution for the semi-permeable outer layer, 0.19 g of PCL and 0.19 g of PVP were added to the 2.98 g of TCM. Just as for the blank solution, 0.005 g of Oil Red O was added for staining purposes. The mixture was intensively stirred for 12 h.

### 2.2. Film Application

A homemade film applicator machine was used to create a thin polymer film on the FEP substrate. The substrate itself was fixed in place using a vacuum table. A 100 µL of each solution was placed at the beginning of the substrate using a pipette and spread with a 25 µm gap baker applicator moving at 20 mm/s. Due to the high viscosity of casted solutions and the high evaporation rate of the TCM, the wet film behind the baker applicator dried almost immediately, preventing the solution from beading on a non-wettable fluorinated substrate. A photo and general description of our home-build film application unit can be found in the Appendix A.

### 2.3. Multilayered Films for Complex-Structured Particles

For complex structure particles, it is necessary to create a master film containing several layers with distinct properties. For example, a particle may consist of a pure PCL baser layer, a cargo-loaded intermediate PCL layer, and a semi-permeable outer PCL layer. At its core, each layer is a thin polymer film formed on a separate FEP substrate. To combine those layers in a single master film, a lamination technique was utilized. Films were heat-fused together in the right order using a commercial laminator machine at 65 °C and a feed rate of 3 mm/s.

### 2.4. Laser Cutting

To cut a master film into a defined size blank, a high-precision laser cutter was used. Substrate movement in the XY plane with high accuracy (±1 µm) was achieved by a pair of high-precision linear stages made by STANDA (8MT167M-25BS1 and 8MTL1301-170-LEn1-100). The linear velocity of the substrate was 5 mm/s. Cutting was performed using Cobolt Tor XS pulsed laser (532 nm, 50 µJ, 1 kHz pulse repetition rate, 1.9 ns pulse duration). A laser beam was expanded by a pair of lenses to fill the back aperture of the focusing lens (LOMO micro-objective, ×3.5 NA = 0.1, estimated spot size 6.5 µm, $spot\approx 1.22\frac{\lambda }{NA}$) for diffraction-limited performances.

In our setup, the laser spot remains stationary, and the sample moves in the XY plane by utilizing motorized linear stages. The repetition rate of the used laser is 1 kHz. Given the theoretical focused spot size of 6.5 microns, the maximum allowed linear speed is computed to be 6.5 mm/s. If this speed is exceeded, the continuous cutting line will be transformed into a linear array of holes (perforation) as the sample will have time to move distances greater than the beam diameter during the time between laser pulses. For thin films (less than 4 microns), one laser pass is enough, but for thicker ones (about 12 microns and above), 3–5 passes are required.

As NS laser pulses were utilized, there is a much larger so-called “heat-affected” zone compared to femtosecond laser sources. PCL is a low-melting-point polymer (60C), and cutting it with a focused laser beam (diffraction-limited spot size of 6.5 microns) usually resulted in many defects in the form of small bridges between adjacent blanks. To avoid these defects, we intentionally defocused our optical system to artificially increase the laser spot size up to 15–20 µm and thus increase the cutting width.

### 2.5. Heat Application

To finalize the particle-manufacturing procedure, the substrate with cut PCL blanks was placed into the accurate oven (Stegler VAC-24, ±0.1 °C) at 90 °C for 3 min for melting and beading. The temperature inside the oven was deliberately set above the typical melting point of the PCL (60 °C) to reduce its viscosity and speed up the beading process.

### 2.6. Elution Kinetics Measurement

The monitoring of cargo elution was assessed by placing the known amount of particles on the substrate into the 1 mL of saline (0.9% NaCl in water) at 37 °C. Each day, the substrate was moved into the fresh volume of the saline. This cycle was repeated for one week. To measure the absolute amount of Methylene Blue eluted each day, an absorption spectrum of 150 µL of the corresponding saline was measured by the TECAN Infinite^®^ 200 Pro M Nano Plus reader. Using the calibration curve, the absorption value at 664 nm was transformed into the absolute amount of the Methylene Blue eluted each day.

### 2.7. Surface Inspection of the Particles

Visual inspection of samples was conducted using an optical transmission microscope Olympus CX-33 and scanning electron microscope (SEM) TESCAN Vega 3.

## 3. Results

### 3.1. General Description of Proposed Manufacturing Method

The described process involves applying a single layer or multiple layers of a master film onto the low-surface-energy substrate. The number of layers, their arrangement, and their properties collectively determine the internal structure and function of the capsules. Through precise, rapid, and deterministic laser cutting, an array of uniformly sized blanks can be generated. By utilizing low-melting-point polymers like polycaprolactone (PCL), a relatively modest increase in temperature, up to 60+ °C, causes the polymer to melt. This enables surface tension forces to rapidly shape the particles into semi-spherical and other forms, depending on their internal composition and the initial shape of the blank. This approach facilitates the efficient and precise manufacturing of various particle types, including single-polymer composites, biopolymers, core-shell structures, and more. Furthermore, the elution profile of the cargo can be finely tuned by adjusting the properties of the outer layer of the master film. The described approach holds the potential to unlock new opportunities in highly reproducible, predictable, and customizable particle manufacturing, suitable for applications ranging from controlled drug delivery to calibration purposes. For a visual representation of the proposed approach for heat-induced particle synthesis on low-surface-energy templates, please refer to Figure 1.

### 3.2. Beading of the PCL Blanks on the FEP Substrate under the Heat

To validate the proposed idea for particle manufacturing, a thin master film of PCL was created on the FEP substrate following the procedures outlined in the second section of the Methods. This master film was then laser-cut into blanks measuring 70 × 70 µm, with a measured thickness of approximately 4 µm. Figure 2a displays a time-lapse sequence illustrating the application of heat and the corresponding beading of PCL blanks. As the substrate temperature approached 60 °C, the blanks began to take on a rounded shape due to melting. A further increase in temperature significantly reduced the viscosity of the PCL, facilitating efficient beading on the non-wettable FEP substrate. When the temperature reached 90 °C, all blanks were transformed into beads with flat bottoms. By adjusting the cutting parameters, such as the dimensions of the blanks, it was possible to produce beads of varying sizes. This is demonstrated in Figure 2b, which showcases beads generated from blanks measuring 25 × 25 µm, 40 × 40 µm, and 70 × 70 µm, respectively.

### 3.3. The Influence of Crystalline Cargo on a Beading Process

The addition of a crystalline component (Methylene Blue) to the polymer is expected to alter the kinetics of bead formation and their geometry. To investigate this effect, a series of PCL solutions containing 5%, 10%, and 15 wt.% of Methylene Blue crystals relative to the mass of PCL were prepared following the procedures outlined in the first section of the Methods. These solutions were then applied to the FEP substrate, as described in the second section, and laser-cut into 70 × 70 µm blanks. SEM analysis of these blanks, as presented in Figure 3a, revealed that they all had a consistent thickness of approximately 4.5 µm, regardless of the Methylene Blue content. This observation can be attributed to the fact that the initial PCL solution, before the addition of Methylene Blue, had sufficient viscosity to ensure a proper application process. Consequently, the addition of cargo microcrystals, which thickens the solution, resulted in a practically negligible change in the thickness of the resulting film [26]. Upon the application of heat, a noticeable difference in bead geometry was observed in relation to the amount of added cargo, as shown in Figure 3b. As the quantity of added cargo increased, it became more evident that the melted polymer tended to interact more with the wettable cargo crystals and less with the non-wettable FEP substrate, causing it to lose its ability to form round beads. Figure 3c illustrates a schematic representation of this phenomenon.

### 3.4. Multilayered Particles

Emanation from single-layer blanks enables the production of simple polymeric particles. The proposed approach is also capable of going over sophisticated layered structures that then result in “core-shell”-like particles. Although each polymer forming a layer has different characteristics, one can manipulate the properties of each layer in the pre-cut film. Hence, it exerts control over various aspects of particle behavior, including elution dynamics and hydrophilicity. To demonstrate this capability, two master films were prepared in accordance with the procedures outlined in the third section of the Methods. The first master film was composed of the following layers (from bottom to top): a blank PCL layer, a PCL layer loaded with Methylene Blue crystals (10 wt.%), and another blank PCL layer. The intent was to create a core containing cargo surrounded by a protective shell that would impede the elution process until it either biodegrades or is damaged by physical stimuli. Figure 4a displays 70 × 70 µm blanks created through laser cutting, while Figure 4b presents the same blanks after undergoing heat treatment at 90 °C for 3 min.

The second master film was constructed with a blank PCL layer at the bottom, an intermediate PCL layer loaded with Methylene Blue (10 wt.%), and an upper layer comprising blended polymers (PCL:PVP mass ratio at 1:1). Since PVP is water-soluble and immiscible with PCL, it can dissolve upon contact with water, creating physical pathways through which cargo can elute into the environment. In Figure 5a,b, we observe 70 × 70 µm blanks from this film before and after heat treatment (90 °C for 3 min). Although the high melting point of PVP (180–200 °C) prevents it from melting at the specified temperature, resulting in minimal visual differences before and after heat treatment at the top of the capsules, the melted PCL effectively seals the cargo inside.

### 3.5. The Influence of Cargo Content and Particle Structure on Elution Kinetics

To quantitatively assess the elution kinetics in relation to the cargo amount, a series of simple particles composed of PCL doped with 5%, 10%, and 15 wt.% of Methylene Blue (MB) were created using the proposed approach. Figure 6a displays the day-by-day MB release profiles for each set of particles. On the first day, an initial elution burst was observed in every group, and this burst was directly proportional to the amount of loaded Methylene Blue. However, SEM analysis of the particle’s morphology before and after incubation in saline, as shown in Figure 6b, did not reveal any significant changes in surface quality. This can be explained as follows: it was observed that some amount of Methylene Blue dissolves in chloroform during the manufacturing process. As a result, a certain portion of the cargo exists in the form of large crystals surrounded by the polymer matrix, while another part consists of water-soluble molecules trapped within the polymer chains. Upon contact with water, these entrapped molecules rapidly escape from the capsules, likely accounting for the substantial substance release observed on the first day, without significant erosion of the capsule surface (Figure 6b). The remaining cargo remains entirely entrapped within the capsules, with no further elution over the subsequent week. The proposed capsule manufacturing method enables precise control over the number of particles produced on the substrate. This simplifies the estimation of the average cargo content in a single particle without the need for sophisticated or less accurate concentration estimation methods. To determine the average cargo content, 4000 particles with varying amounts of cargo (5%, 10%, and 15 wt.%) were immersed in 200 µL of distilled water. Subsequently, 100 µL of ethyl acetate was added to the mixture to dissolve the polymer and extract the Methylene Blue into the aqueous phase. Carefully collecting 150 µL of the water from the bottom of the vial, spectroscopic analysis was performed to estimate the absolute amount of the dye. Figure 6c illustrates the average cargo per capsule for different amounts of Methylene Blue added to the initial polymer solution. The loading capacities were found to be 0.56 ± 0.1 ng/capsule, 1.02 ± 0.12 ng/capsule, and 1.57 ± 0.1 ng/capsule, which closely corresponded to the ratios of 5%, 10%, and 15 wt.% of the dye, respectively.

In Figure 7a,b, daily elution profiles and SEM images of particle morphology before and after one week of incubation in saline are presented for core-shell structured particles with an impermeable PCL shell. It is evident that the addition of the impermeable shell effectively blocks the elution of the cargo. This property could be harnessed for on-demand drug release triggered by external physical stimuli. Achieving such behavior could involve incorporating specific sensitizers into the shell, such as magnetic or gold nanoparticles, which can convert the energy from alternating magnetic fields or photons into heat, therefore rupturing the shell for the on-demand therapy [27,28,29].

To elaborate the formulation with prolonged drug release, we examined test particles with a semi-permeable PCL-PVP shell. As observed in Figure 8a, the addition of a permeable shell reduced the initial day’s elution by approximately 10% compared to plain composite PCL particles containing Methylene Blue crystals. Unlike simple PCL composite particles, there was continuous elution of Methylene Blue for up to 4 days. However, the elution kinetics do not reflect normal release decay curves (Figure 8a) due to a noticeable difference in the amount of cargo eluted on the first day compared to the rest of the period. By this experiment, we demonstrate the capacity to alter the release profile by modifying the properties of the shell. Figure 8b provides a visual representation of the effect of PVP addition on the capsule’s surface before and after incubation in saline.

### 3.6. Bi-Polymer (Janus or Asymmetrical) Particles

To highlight the versatility of using polymers other than the low-melting-point PCL, we created a set of asymmetric or Janus particles [30]. These particles were fabricated from a two-layer master film consisting of PLA and PCL layers (Figure 9). At 90 °C, the PLA base of the particles did not melt, but the PCL top layer completely melted and, due to differences in density, floated to the top. The combination of unique discrete features (materials) within the single anisotropic particle significantly expands its functionality. Particles of this kind may be in demand not only in the field of targeted drug delivery but also in the field of biotechnology, micro-manipulation, and other industries where precise control over the properties of the environment or the number of substances involved in the reaction is necessary [31,32,33]. The authors suggest that the proposed method can significantly facilitate and speed up the procedure for manufacturing asymmetric particles based on biodegradable thermoplastic polymers, which in turn will invariably lead to their wider and accelerated implementation for solving various problems.

## 4. Discussion

In this study, we have presented a straightforward yet highly effective approach for producing an array of particles with predefined structures and properties. The throughput of this method relies entirely on the linear speed of the translation stages and the pulse repetition rate of the laser. In our case, the production rate was constrained by the 1 kHz repetition rate of the laser system. To create 4000 particles, our system required over two minutes to cut the master film. It is worth noting that current industrial lasers can reach tens of kHz or even higher repetition rates, potentially speeding up production by an order of magnitude or more. The authors of this work envision significant potential for scaling this technique, as it does not require specialized templates or specific environmental conditions. This process can be carried out continuously using roll-to-roll and slot-die coating approaches, either with a single-layer master film or multiple layers.

The proposed approach is highly robust because the manufacturing of a simple flat film on a web substrate is a well-established and predictable method for creating thin films with minimal defects. By combining several layers with predefined properties, functionally distinct particles can be easily generated. Additionally, the laser-cutting parameters control the size of the resulting particles, making it possible to create a precise quantity of particles with different sizes and predefined ratios on a single substrate. This capability could prove useful for calibration purposes or for studying the biodistribution of these particles based on their size [34,35].

The minimal adhesion between the low-energy FEP substrate and the master film has to be considered. While cutting this film with a pulsed laser, micro-explosions occur due to the ablation phenomenon, attempting to dislodge the blanks from the substrate. In our work, we were unable to reduce the blank size below 25 × 25 µm, as doing so resulted in a high number of defective blanks (turned, flipped, or displaced). From this perspective, one can look forward to exploiting this technique for particle formation to extend over different polymers, such as PLA, PLGA, or other commonly used biopolymers for medical applications. However, their relatively high melting points (180+ °C for PLA and 200+ °C for PLGA) raise concerns about the thermal stability of the cargo.

It appears to the authors that at some sizes and thicknesses of blanks, the transformation from square to hemispherical particles will break down. Within this paper, we tested the thicknesses from 4 to 12 microns and blank sizes from 25 to 70 microns. If there is no cargo added to the polymer, these thicknesses and sizes result in hemispherical particles after the heat treatment.

The particular laser wavelength for cutting is chosen by the intrinsic absorption (linear absorption) line either of the plane polymer or the additives. Usually, plane polymers absorb UV, with rare exceptions. However, UV optics are quite expensive and complicated. To shift the absorption line to the convenient visible or NIR region, some staining or pigmentation of the polymer is required. The authors do not see any mechanism of how the wavelength of light by itself can alter the final structure of particles. However, the addition of staining agents or pigments can influence the formation of the beads by altering the surface tension of the polymer.

## 5. Conclusions

This work described a scalable method for fabricating polymer particles on low-surface-energy substrates. The proposed method allows the fabrication of particles of different structures, ranging from simple composite particles to multi-polymer core-shell structures. Different release kinetics of the packaged substances can be achieved by varying the properties of the top layer of the particles. For example, a weakly permeable polycaprolactone shell practically stops the release of the substance from the capsule core. At the same time, a semi-permeable shell made of a mixture of polycaprolactone and polyvinylpyrrolidone allows the packaged substance to be released into the environment within the first few days. The application of laser-cutting technology allows the quick and accurate obtaining of blanks for particles of the required size or combinations of sizes in the required proportions, which can be useful in the creation of multi-size drug depots since it is obvious that particles of different sizes will have slightly different release kinetics. The main step of the technology is the creation of a thin polymer film on the non-wetting surface of fluorinated ethylene propylene (FEP), which is possible when a certain viscosity of the polymer solution is reached. This method of film creation can be applied in combination with slot-die coating technologies, which are among the most productive in creating thin films on various surfaces. Using high-speed laser-cutting units (cutting speed > 1 m/s), according to the authors, the proposed method is promising for manufacturing a large number of capsules with small variations in their sizes and properties, which is in demand in modern targeted therapy of various diseases and effective use of drugs.

## Figures and Tables

**Figure 1 nanomaterials-13-02976-f001:**
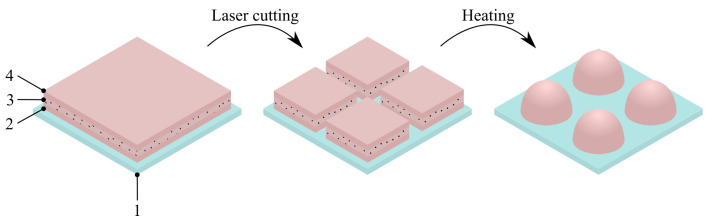
Illustration of the basic principle of capsule manufacturing on low-surface-energy substrate. First, a master film is created on top of the Fluorinated Ethylene Propylene (FEP) substrate 1. The master film can consist of several layers. For example, it can be made out of bottom layer 2, intermediate or cargo layer 3, and top protective layer 4. Next, a precise and computer-guided laser cutting takes place to split the master film into a desired number of blanks with a defined size. Heat is then applied to melt the polymer, thus reducing its viscosity and allowing surface tension forces to make hemispherical or over-shaped particles. The viscosity of the melt and capsule content affects the resulting geometries of the particles.

**Figure 2 nanomaterials-13-02976-f002:**
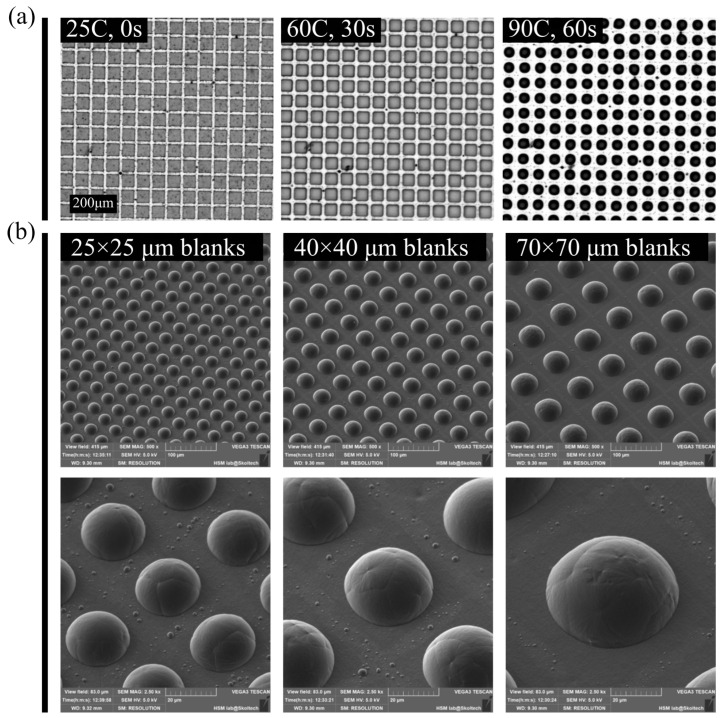
(**a**)—a time-series of frames illustrating the beading process of blanks on the FEP substrate. After passing the melting point, rectangular blanks can be clearly seen under the influence of the surface tension force, which quickly form hemispheres. (**b**)—SEM images of various size particles produced from different-sized blanks. This figure shows the ability to precise control over the size of particles with almost zero diameter variance within each batch.

**Figure 3 nanomaterials-13-02976-f003:**
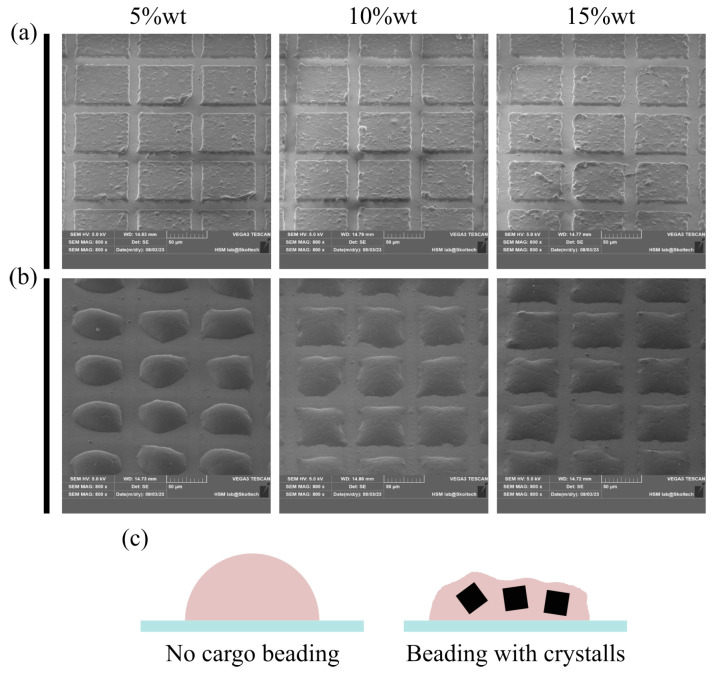
(**a**)—a laser-cut master single-layer PCL film containing 5, 10, and 15 wt.% of Methylene Blue. (**b**)—particles formed after the heat application. The addition of crystalline cargo affects the beading process. (**c**)—illustration of the potential mechanism of reduced beading in the presence of crystalline components. The relatively large surface area of the melted polymer interacts with the surface of cargo crystals, which is apparently wettable by the melt. This interaction interferes with the production of polymer hemispheres on the substrate, thus forming particles of irregular geometry.

**Figure 4 nanomaterials-13-02976-f004:**
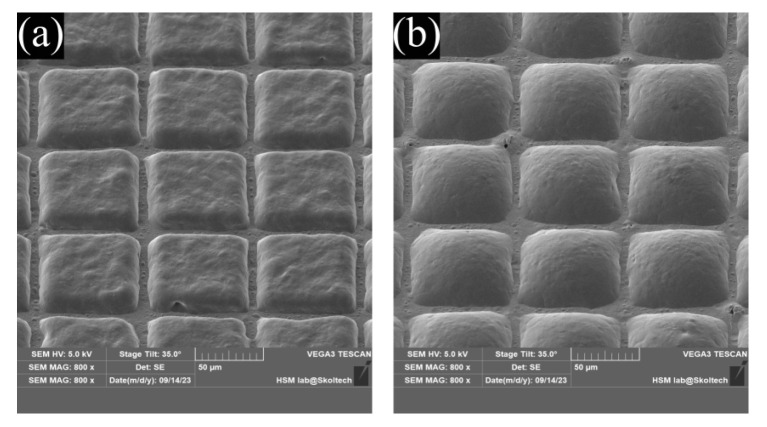
(**a**)—Multilayered master film consisting of plane base PCL layer, cargo carrier PCL layer in the middle, and plane PCL protective layer on top. (**b**)—changes in the shape of capsules after the heat application. Because all layers of the capsule are made of the same polymer, after the melt, there will be no mixing of the layers due to different densities. Along with this, the polymer melt is viscous enough to significantly slow down the process of particles sinking. Based on this, we can propose that, in this way, it was possible to create a capsule of the core-shell type, where the “core” means a region of the capsule with a significantly higher cargo density compared to the “shell”.

**Figure 5 nanomaterials-13-02976-f005:**
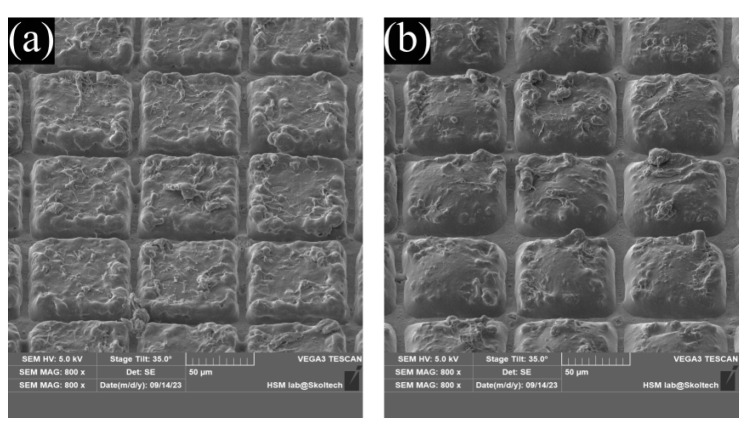
(**a**)—Multilayered master film consisting of plane base PCL layer, cargo carrier PCL layer in the middle, and blended polymers (PCL-PVP, 1:1) protective layer on top. (**b**)—changes in the shape of capsules after the heat application. It is clear from the image that there is a complete melt of PCL components of the master film but no visual changes in PVP clusters due to the large difference in melting point of these polymers.

**Figure 6 nanomaterials-13-02976-f006:**
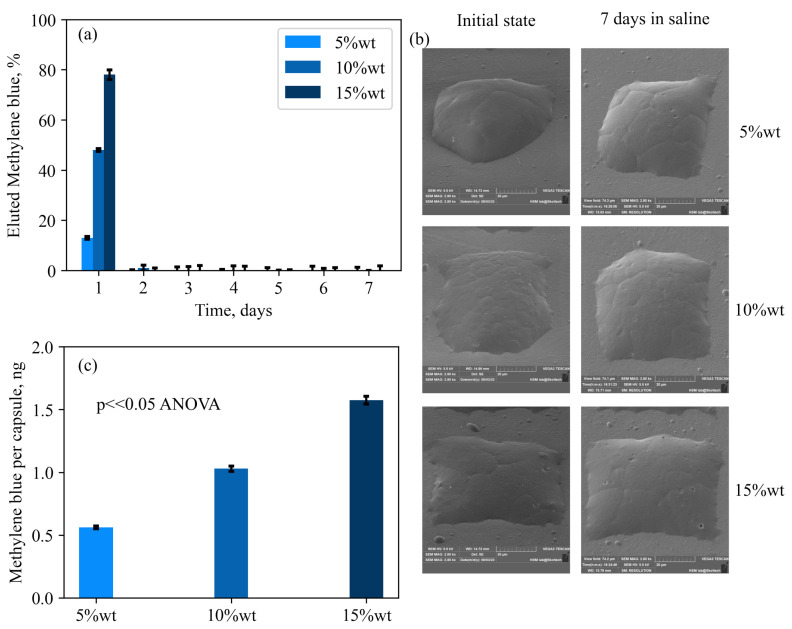
(**a**)—Daily elution profile for simple PCL composite particles with different amounts of loaded Methylene Blue dye. It is clear from the picture that there is a first-day release burst proportional to the amount of cargo. Almost no elution was detected at the reset of the one-week period. (**b**)—difference in the surface morphology of the above-mentioned capsules before and after the incubation reveals no major difference besides a small number of pores or crates. (**c**)—average payload per capsule for different amounts of added dye. The ANOVA analysis shows a *p*-value much less than 0.05, which is strong evidence of the statistical difference between the cargo amount per capsule in each group.

**Figure 7 nanomaterials-13-02976-f007:**
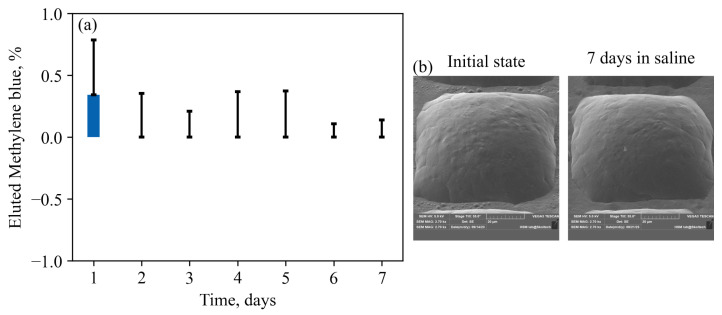
(**a**)—elution profile for a core-shell particles. Practically no release was detected besides a small leak of about 0.35% on the first day. A large amount of errors is due to an extremely small concentration of Methylene Blue in the solution, which is hard to distinguish from pure saline. (**b**)—Typical surface of a core-shell particle before and after the incubation in the saline. No noticeable difference was visualized.

**Figure 8 nanomaterials-13-02976-f008:**
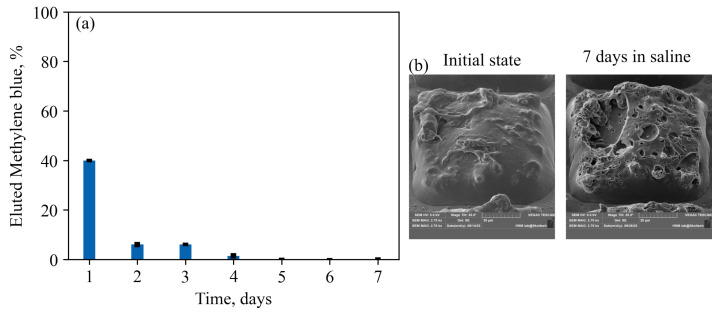
(**a**)—elution profile for core-shell particles with a permeable outer layer made of blended polymers PCL-PVP (1:1). The data show a decrease in the release of the substance on the first day and a systematic release of a small amount of the substance over four days. (**b**)—SEM images of the capsule’s surface before and after the incubation in the saline. It is clearly seen from this figure that the water-soluble clusters of PVP leave behind a large number of pores and craters through which a small amount of substance can diffuse, thus allowing a prolonged elution compared to plane composite particles.

**Figure 9 nanomaterials-13-02976-f009:**
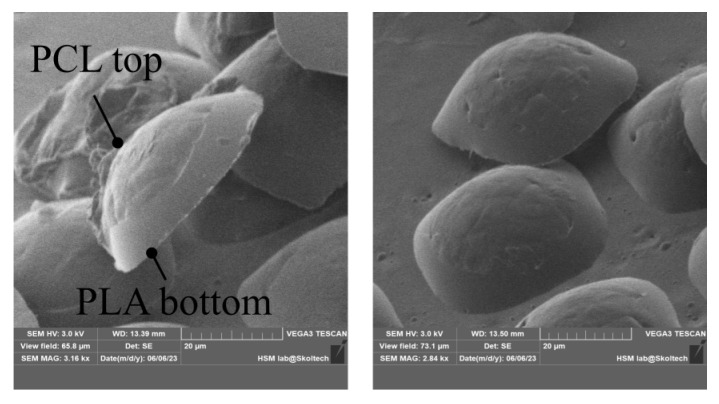
Particles made from two distinct polymers with varying crystallinity, degradation rates, and numerous other properties. This further underscores the flexibility and simplicity of the proposed method for producing particles with well-defined structures and properties while maintaining minimal defects.

## Data Availability

Data are contained within the article.

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
