# Peer review of "Template-Free Manufacturing of Defined Structure and Size Polymeric Microparticles"

_nanomaterials, 2023, doi:10.3390/nano13222976_

Round 1

Reviewer 1 Report

Comments and Suggestions for Authors

In this manuscript, the authors present a novel and scalable approach for the manufacturing of complex-structured polymeric microparticles designed for next-generation medicine, primarily for targeted drug delivery. By introducing a method that focuses on the deposition of polymer layers on a low-surface-energy substrate and subsequent precise laser cutting, the authors address prevalent challenges in producing microparticles with desired properties. Specifically, this work promises advancements in microparticle production without relying on micro-patterned templates, while also ensuring control over particle size distribution and cargo release kinetics. The subject is captivating and holds immense potential for medical applications. However, I recommend addressing several points before considering the manuscript for publication.

1. It's imperative to expand abbreviations, like PVP, to their full form upon their initial mention. Please refer to Page 13 as an example.

2. The scale bar of Figure 2a should be added. 

3. The statistical testing method should be added in Figure 6c.

4. Some time points (e.g. 6h, 12h) of elution profile should be added in Figure 6a.

6. In Figure 8a, what is the sample size?

5. I suggest citing pioneering work in the introduction, specifically referencing the following sources: Advanced Functional Materials, 2212952.

Author Response

Dear Editor,

The authors would like to thank all reviewers for their valuable and important comments. We have revised our manuscript accordingly to them. Please, find below the list of reviewers' comments and our responses. All changes done within the manuscript are highlighted in blue.

1. It's imperative to expand abbreviations, like PVP, to their full form upon their initial mention. Please refer to Page 13 as an example.

Response: Thank you for noticing this. We detailed all abbreviations once they first appeared

2. The scale bar of Figure 2a should be added.

Response: Thank you for pointing out missing scale bar. We added it.

3. The statistical testing method should be added in Figure 6c.

Response: Thanks for pointing out the missing statistical analysis in this figure. We performed an ANOVA test to confirm that there is a statistical difference between the average loading per capsule for three groups of particles. Results of ANOVA test are added in revised version at description of the Figure 6.

4. Some time points (e.g. 6h, 12h) of elution profile should be added in Figure 6a.

Response: Within this work, we focus on the particles creation for long-term (over days and possibly weeks) drug elution and therapy. The main idea was to minimize the initial drug burst in first 24 hours and spread out the release of the substance over time, ideally for a week or more. According to that concept, we were not interested in the eluted amount of the substance released within first hour. Short-term drug delivery micro-systems are well established and studied. They can be used for drug delivery in a time-frame of minutes and hours. In the introduction section of our paper we added a paragraph, describing the already well elaborated hydrogel-based drug delivery systems used for short-term drug dosage and the demand for highly efficient and simple manufacturing methods of long-term drug elution devices.

6. In Figure 8a, what is the sample size?

Response: Sample size was approx. 4000 particles plus or minus 5% per group as was added in Sect. 3.5.

5. I suggest citing pioneering work in the introduction, specifically referencing the following sources: Advanced Functional Materials, 2212952.

Response: Thanks for pointing out this pioneering work. We referenced this work to the introduction section of our manuscript.

Reviewer 2 Report

Comments and Suggestions for Authors

In this study, the authors introduce a scalable, template-free method for manufacturing polymeric microparticles with precise structures and sizes. By depositing polymer layers on a low surface energy substrate and using laser cutting followed by heat application, the method produces discrete particles with hemispherical and rectangular geometries. These microparticles are promising for targeted drug delivery and other advanced medical applications. 

The article stands out for its clarity and well-structured presentation, making it easily comprehensible for readers. To further enhance the manuscript, authors should consider addressing the following comments.

1) Could the author provide the pulsed width of the laser used? 

2) Did the author identify any specific size and thickness thresholds for the cut blank rectangular structures and film thickness, beyond which it becomes challenging to achieve a hemispherical shape?

3) Your figures are valuable visual aids. To enhance their accessibility, kindly provide the scale for Fig. 2a.

4) If the laser spot size is 6.5 microns, then how the cutting path becomes 20-30 microns in Fig 3. Kindly justify.

5) Typo error in Fig 3a caption. “It is clear the the addition …. process.” It should be “It is clear that the addition…. process.”

6) Do the final structure (so-called particles) changes with the change of laser wavelength..? can the author comment on this aspect? 

7) Understanding the laser scan speed is crucial in comprehending the cutting process for polymeric films. Could authors provide details on the laser scan speed? Additionally, does the laser make multiple passes or just one pass to cut through the single-layered and multilayered films?

8) How does increasing the repetition rate significantly enhance particle production? Does the author want to say that throughput depends on laser “scan rate” rather than “repetition rate”? 

9) Your manuscript is thorough; however, it would be enhanced by the addition of a dedicated conclusion section summarizing the key findings and their implications.

10) To enhance reader understanding, authors need to provide more explanations of terminology such as 'Blank' and 'cargo' and others. This would cater to a broader readership.

11) The authors are encouraged to offer a more comprehensive explanation of the solution preparation and film application sections, covering all systematic processes and parameters employed in their study.

12)I find the homemade film applicator machine described in your study intriguing. Providing a picture or schematic, along with a clear explanation of its functioning for film preparation, would greatly benefit readers interested in replicating your methodology.

13) I noticed a typographical error in section 2.2. 'Begging' should be corrected to 'beginning' for accuracy."

14) In the laser cutting section, there seems to be a unit discrepancy for ±1. Could you please clarify if it is in millimeters or microns? 

15) Could you provide a justification or explanation to clarify how a 3.5X objective lens results in a 6.5-micron spot size?

Comments on the Quality of English Language

NA

Author Response

Dear Editor,

The authors would like to thank all reviewers for their valuable and important comments. We have revised our manuscript accordingly to them. Please, find below the list of reviewers' comments and our responses. All changes done within the manuscript are highlighted in blue.

1) Could the author provide the pulsed width of the laser used? 
Response: The pulse width of used laser is 1.9 ns. We added this info into the materials and methods section in revised version of our manuscript.

2) Did the author identify any specific size and thickness thresholds for the cut blank rectangular structures and film thickness, beyond which it becomes challenging to achieve a hemispherical shape?
Response: We do not specifically test the limits in terms of blank size and film thickness. Nevertheless, it is appeared to authors that at some sizes and thicknesses the transformation from square blank to hemispherical particles will breaks down. Within this paper, we tested the thicknesses from to 4 to 12 microns and blank sizes from 25 to 70 microns. In case of no cargo added to the polymer, these thicknesses and sizes resulted in semispherical particles after the heat treatment.
We added some information about this to the discussion section of the paper.

3) Your figures are valuable visual aids. To enhance their accessibility, kindly provide the scale for Fig. 2a
Response: Thank you for pointing out the lack of scale bar. We added it on Fig 2a.

4) If the laser spot size is 6.5 microns, then how the cutting path becomes 20-30 microns in Fig 3. Kindly justify.
Response: As we using ns laser pulses, there is a much larger so-called “heat-affected” zone, compared to femtosecond laser source. PCL is a low melting point polymer (60C) and cutting it with focused laser beam (diffraction limited spot size of 6.5 microns) usually resulted in many defects in a form or small bridges between adjacent blanks. In order to avoid these defects we intentionally defocus our optical system to artificially increase the laser spot size and increase the cutting-width. For convenience of readers we
clarified the difference between the diffraction limited spot size and cutting width in the laser cutting section of the paper (Sect 2.4) .

5) Typo error in Fig 3a caption. “It is clear the the addition …. process.” It should be “It is clear that the addition…. process.”
Response: Thank you for pointing out this typo. We corrected it.

6) Do the final structure (so-called particles) changes with the change of laser wavelength..? can the author comment on this aspect?
Response: Particular laser wavelength is choosing by the intrinsic absorption line either of the plane polymer or the additives. Usually, plane polymers absorbing in UV, with rare exceptions. But UV optics is quite expensive and complicated. To shift the absorption line to the convenient visible or NIR region, some staining or pigmentation of the polymer is required. Authors do not see any mechanism of how the wavelength of light by itself can altering the final structure. But the addition of staining agents or pigments can influence the formation of the beads by altering the surface tension of the polymer. This information was added to the discussion section of our paper.

7) Understanding the laser scan speed is crucial in comprehending the cutting process for polymeric films. Could authors provide details on the laser scan speed? Additionally, does the laser make multiple passes or just one pass to cut through the single-layered and multilayered films?
Response: In our setup laser spot remains stationary and sample moves in XY plane by utilizing motorized linear stages. The repetition rate of used laser is 1kHz. Given the theoretical focused spot size of 6.5 microns the max allowed linear speed is computed to be 6.5 mm/s (speed = distance/time). If this speed is exceeded, the continuous cutting line will be transformed into a linear array of holes (perforation) as the sample will have time to move distances greater than the beam diameter during the time between laser pulses. For thin films (approx. 4 microns) one laser pass in enough, but for thicker ones (about 12 microns) we used 3-5 passes. For the reader’s convenience, we expanded the corresponding section of our revised manuscript to include these details about cutting regimes (sect. 2.4).

8) How does increasing the repetition rate significantly enhance particle production? Does the author want to say that throughput depends on laser “scan rate” rather than “repetition rate”? 
Response: The repetition rate of used laser is 1kHz. Given the theoretical focused spot size of 6.5 microns the max allowed linear speed is computed to be 6.5 mm/s (speed = distance/time). If this speed is exceeded no continuous cutting will occur but rather the perforation takes place. By increasing the laser pulse repetition rate the max-allowed sample velocity (or “laser scan speed”) can be increased thus busting the capsule manufacturing capacity per unit time. To clarify the terminology, by the “laser pulse repetition rate” we mean the time between two adjacent laser pulses, and by the “laser scan rate” or “cutting-speed” authors means the velocity at which sample moves with respect to the stationary laser spot or vice-versa. This information was included in the conclusion section.

9) Your manuscript is thorough; however, it would be enhanced by the addition of a dedicated conclusion section summarizing the key findings and their implications.
Response: Journal template is suggesting combining the conclusion and discussion sections into a one big block, but for reader’s convenience we arrange a separate conclusive paragraph as section summarizing the main results and implication of our paper.

10) To enhance reader understanding, authors need to provide more explanations of terminology such as 'Blank' and 'cargo' and others. This would cater to a broader readership.
Response: Thank you for this comment. We added description of some specific terms throughout the article.

11) The authors are encouraged to offer a more comprehensive explanation of the solution preparation and film application sections, covering all systematic processes and parameters employed in their study.

Response: We added the additional info to the solution preparation and film application sections.

12)I find the homemade film applicator machine described in your study intriguing. Providing a picture or schematic, along with a clear explanation of its functioning for film preparation, would greatly benefit readers interested in replicating your methodology.
Response: The technical aspects of basic film application unit is out of the scope of this paper. We include photons and general description of our in-house-build film application unit into the supporting information. Supporting information figure 1.

13) I noticed a typographical error in section 2.2. 'Begging' should be corrected to 'beginning' for accuracy."
Response: Thank you for noticing the typo, we corrected it.

14) In the laser cutting section, there seems to be a unit discrepancy for ±1. Could you please clarify if it is in millimeters or microns? 
Response: Indeed, microns were missed. We added units. Thank you for noticing.

15) Could you provide a justification or explanation to clarify how a 3.5X objective lens results in a 6.5-micron spot size?
Response: 6.5 microns come from the theoretical calculations using Abbe equation spot = 1.22λ/NA,  there NA is numerical aperture of the used lens, and λ is the wavelength of light.

Round 2

Reviewer 2 Report

Comments and Suggestions for Authors

I recommend publishing this journal article as I am pleased to note that all of my comments to the authors have been thoroughly addressed. The manuscript has undergone significant improvement, and I confidently recommend it for publication.